# Incidental Node Metastasis as an Independent Factor of Worse Disease-Free Survival in Patients with Papillary Thyroid Carcinoma

**DOI:** 10.3390/cancers15030943

**Published:** 2023-02-02

**Authors:** Renan Aguera Pinheiro, Ana Kober Leite, Beatriz Godoi Cavalheiro, Evandro Sobroza de Mello, Luiz Paulo Kowalski, Leandro Luongo Matos

**Affiliations:** 1Head and Neck Surgery Department, Instituto do Câncer do Estado de São Paulo, Hospital das Clínicas da Faculdade de Medicina da Universidade de São Paulo, São Paulo 05403-000, Brazil; 2Faculdade Israelita de Ciências da Saúde Albert Einstein, São Paulo 05653-120, Brazil; 3Pathology Department, Instituto do Câncer do Estado de São Paulo, São Paulo 01246-000, Brazil; 4Laboratório de Investigação Médica 14 (LIM14), Faculdade de Medicina da Universidade de São Paulo, São Paulo 01246-903, Brazil; 5Laboratório de Investigação Médica 28 (LIM28), Faculdade de Medicina da Universidade de São Paulo, São Paulo 01246-903, Brazil

**Keywords:** papillary thyroid carcinoma, lymph node dissection, metastasis, incidental findings, prognosis

## Abstract

**Simple Summary:**

Papillary thyroid cancer is treated mainly by thyroidectomy surgery. The surrounding lymph nodes are not usually resected unless it is known before surgery that they are metastatic, and then all adjacent lymph nodes are resected (named the central compartment neck dissection). However, some lymph nodes could be incidentally resected with the thyroid, sometimes containing metastasis, but this does not mean that the central compartment neck dissection was performed properly. This study aimed to test whether these patients with incidental metastatic nodes had higher treatment failure rates. We found that they indeed had higher rates of treatment failure, even when compared to patients with clinically evident central compartment node metastasis that were submitted to proper neck dissection. We suggest that these patients must be closely followed to detect signs of treatment failure early and to provide prompt treatment.

**Abstract:**

Introduction: Papillary thyroid carcinoma (PTC) have high node metastasis rates. Occasionally after thyroidectomy, the pathological report reveals node metastasis unintentionally resected. The present study aimed to evaluate the prognosis of these patients. Methods: A retrospective cohort of patients submitted to thyroidectomy with or without central compartment neck dissection (CCND) due to PTC with a minimum follow-up of five years. Results: A total of 698 patients were included: 320 Nx, 264 pN0-incidental, 37 pN1a-incidental, 32 pN0-CCND and 45 pN1a-CCND. Patients with node metastasis were younger, had larger tumors, higher rates of microscopic extra-thyroidal extension, and angiolymphatic invasion and most received radioiodine therapy. Treatment failure was higher in patients pN1a-incidental and pN1a-CCND (32% and 16%, respectively; *p* < 0.001—Chi-square test). Disease-free survival (DFS) was lower in patients pN1a-incidental compared to patients Nx and pN0-incidental (*p* < 0.001 vs. Nx and pN0-incidental and *p* = 0.005 vs. pN0-CCND) but similar when compared to patients pN1a-CCND (*p* = 0.091)—Log-Rank test. Multivariate analysis demonstrated as independent risk factors: pT4a (HR = 5.524; 95%CI: 1.380–22.113; *p* = 0.016), pN1a-incidental (HR = 3.691; 95%CI: 1.556–8.755; *p* = 0.003), microscopic extra-thyroidal extension (HR = 2.560; 95%CI: 1.303–5.030; *p* = 0.006) and angiolymphatic invasion (HR = 2.240; 95%CI: 1.077–4.510; *p* = 0.030). Conclusion: Patients that were pN1a-incidental were independently associated with lower DFS.

## 1. Introduction

Thyroid cancer is the main endocrine malignancy. Globally the estimated number of new cases in 2020 was about 449,000 and 137,000, respectively, in women and men [1]. Most of them are differentiated thyroid carcinomas originating from the follicular cells, and about 85% of these are papillary thyroid carcinomas (PTC), usually a low-grade malignancy [1,2,3]. Treatment bases involve surgical resection (thyroidectomy and neck dissection, when necessary), radioiodine therapy for high-risk patients and thyroid stimulating hormone (TSH) suppression based on individual risk stratification. Generally, patients have excellent overall survival rates (over 90% in 10 years) [2,3].

One of the major features of PTC is the high prevalence of lymph node metastasis, especially in the central neck compartment, which varies between 20 to 90% and may impact locoregional recurrence [4]. This wide range of incidence remains a controversial field of debate whether elective central compartment neck dissections (CCND) should be performed routinely in these patients once its benefit is not completely clear in terms of survival. Over the years, several studies sought to analyze the risk factors for lymph node metastasis in clinically negative patients. Factors like sex, age, size, location, laterality, multifocality, extrathyroidal extension, and angiolymphatic invasion were identified [4,5]. Some of these factors, along with others and the lymph node status, are considered in the risk stratification system adopted by the American Thyroid Association (ATA) guidelines to predict disease recurrence [3].

Most of the published studies that evaluated disease recurrence had retrospective designs due to the long natural history of the disease. Usually comparing patients submitted to thyroidectomy and CCND, sometimes including patients that had lymph node representation in the pathological examination but did not undergo a systematic CCND, whereas others submitted only to thyroidectomy. However, the fact that after a total thyroidectomy, the pathological report occasionally reveals incidental lymph node resection means those that were unintentionally resected by the surgeon, which is eventually positive for metastatic disease but does not mean that CCND was performed when examining only the pathological report. Therefore, we questioned if patients with incidental lymph node metastasis are a group of patients that could require a specific follow-up and management. This topic is not well explored in the relevant literature.

Based on that, this study evaluated if the incidental lymph node metastasis, identified exclusively by pathological examination, has prognostic implications in the follow-up of patients with PTC, especially in treatment failure and disease-free survival in a real-world scenario, comparing them with other groups of patients, including those with known lymph node metastasis in the central neck compartment before or during the surgery.

## 2. Materials and Methods

### 2.1. Ethics

The study was approved by the Institutional Review Board of the Hospital das Clínicas da Faculdade de Medicina da Universidade de São Paulo (HCFMUSP; protocol number: 3.988.939; CAAE: 32884214.5.0000.0065). The waiver of the informed consent statement due to the retrospective design was also approved.

### 2.2. Study Design

A retrospective cohort study was conducted with consecutive patients with PTC who were surgically treated with curative intent and followed by the Head and Neck Surgery Service of the Instituto do Câncer do Estado de São Paulo (ICESP) of the HCFMUSP from January 2009 to December 2015, aiming for at least a minimum of five years of follow-up.

The patients included were all older than 18 and had submitted to partial thyroidectomy and totalization within three months from initial surgery, total thyroidectomy, and total thyroidectomy with CCND.

Those patients with synchronous follicular thyroid carcinoma, non-invasive follicular thyroid neoplasm with papillary-like nuclear features (NIFTP), other histologic malignant neoplasms, previously submitted to central neck compartment surgical procedure for any reason, concomitant lateral neck dissection at the first surgery, known distant metastasis before initial surgical treatment or macroscopic residual disease in the first surgical treatment were excluded from the study.

The patients were divided into five groups: Group 1: Total thyroidectomy without lymph node resection; Group 2: Total thyroidectomy with incidental lymph node resection but no lymph node metastasis; Group 3: Total thyroidectomy with incidental lymph node resection and lymph node metastasis; Group 4: Total thyroidectomy with CCND (elective) but no lymph node metastasis; and Group 5: Total thyroidectomy with CCND (elective or therapeutic) and with any lymph node metastasis.

### 2.3. Data Collection

Demographic, clinical, surgical, pathological and follow-up data were recorded in the Research Electronic Data Capture software (REDCap© 11.2.5-2022 Vanderbilt University) hosted by HCFMUSP. These data were obtained through electronic medical records, pathological, laboratory tests and imaging reports. When necessary, an experienced pathologist (E.S.M.) performed the pathologic reexamination. It is noticeable that all pathological reports follow the College of American Cancer Reporting Protocols, also reporting the absent variables.

Thyroglobulin and anti-thyroglobulin levels were assessed on four occasions: (a) stimulated thyroglobulin during whole-body scintigraphy (WBS) only or associated with radioiodine therapy immediately after surgery; (b) one year after treatment; (c) last laboratory control immediately before recurrence identification and (d) in the last medical appointment. Except for the stimulated thyroglobulin levels, the other measures were noted under adequate TSH suppression; otherwise, they were considered missing. The tests were obtained through chemiluminescence immunometric assays, and their functional sensitivity was <0.2 ng/mL for thyroglobulin and <4 IU/mL for anti-thyroglobulin. The anti-thyroglobulin was recorded as a continuous variable but analyzed as a dichotomous variable (negative < 4 IU/mL or positive ≥ 4 IU/mL).

The WBS indications were diagnostic WBS or during the radioiodine therapy period. The radioiodine therapy indications were remnant ablation, adjuvant therapy and therapy of persistent disease. This information was not always available in the medical records, and when it was not explicit, we considered the iodine-131 dose to separate remnant ablation (≤150 mCi) from adjuvant therapy (>150 mCi) according to the 2015 American Thyroid Association Guidelines [3].

The Cancer Staging Manual (8th edition) [6] was used for the purpose of TNM and Group staging. Moreover, the 2015 American Thyroid Association Guidelines [3] were also adopted for the recurrence risk stratification immediately after surgery and for the response-to-therapy categories one year after treatment (surgery alone or surgery and radioiodine therapy).

### 2.4. Central Compartment Neck Dissection Definition

The American Head and Neck Society Consensus Statement [7] defines the central neck compartment comprehends the levels VI and VII lymph nodes. The first level is delimited by the hyoid bone, sternal notch, medial aspect of carotid arteries, prevertebral fascia, and the superficial layer of the deep cervical fascia. Level VII lays caudal to level VI; its lower limit is the brachiocephalic trunk, where it crosses the trachea on the right and the corresponding axial plane on the left. These level nodes are further subdivided based on their location. The pretracheal, paratracheal (left and right), and prelaryngeal lymph nodes are mostly involved.

Therefore, the systematic CCND in the treatment of the PTC determines the resection of the prelaryngeal, pretracheal and paratracheal lymph nodes, which may be unilateral or bilateral. It was only when the CCND was performed, comprehending all these groups of lymph nodes that the patients were included in Groups 4 or 5.

### 2.5. Incidental Lymph Nodes and Incidental Lymph Node Metastasis Definition

The unaware lymph node resection defined the incidental lymph nodes in this study by the surgeon that was found exclusively in the pathological examination or those resected by opportunity (e.g., prelaryngeal lymph nodes). However, these lymph nodes were not clinically suspicious for the metastatic disease before surgery or intraoperatively. The incidental lymph node metastasis was those patients with clinically negative nodes found to be positive after the final pathological examination (cN0 discovered to be pN1a). These patients had their surgical records and or their preoperative ultrasound reviewed to ensure this information. Lymph nodes were considered suspicious for metastatic disease in the presence of calcifications, loss of echogenic hilum, hyperechogenicity, abnormal shape, or increased vascularity in the preoperative ultrasound and intraoperatively based on the size, form, color, consistency, tissue invasion and mobility.

### 2.6. Statistical Analyses

The values obtained by the parametric quantitative variables were organized and described by mean, standard deviation, and non-parametric data by median and inter-quantile range. Relative and absolute frequencies were used for qualitative data. Fisher’s exact test or chi-square test was used in the comparison of frequencies between groups. The comparisons between continuous variables and the groups were evaluated with Mann–Whitney U test and the Kruskal–Wallis test, followed by Dunn’s auxiliary test, due to no parametricity being evaluated by Kolmogorov–Smirnov test. The Kaplan–Meier method was used in the survival analyses, and the Log-Rank test in the curve’s comparisons between groups. The Cox proportional-hazards regression model was used to investigate associations with disease-free survival (DFS) in univariate and multivariate analyses for those variables with *p* < 0.10 at univariate analyses. The method of independent proportions was applied during power analyses. The descriptive analyses were performed using the R statistical software (R version 4.1.2-RStudio 1.4) and the statistical analyses and comparisons with the SPSS^®^ 29.0 software (SPSS^®^ Inc. Chicago, IL, USA.). The significance level adopted was 5% (*p* ≤ 0.05) as the probability of making an α or type I error.

The disease-free survival interval was considered as the time between the day of the surgery and the date of the earliest treatment failure identification or the day of the last medical appointment for those without this outcome. Treatment failure was regarded as disease persistence or disease recurrence (locoregional or distant) when it occurred within or after six months of follow-up, respectively.

## 3. Results

### 3.1. Cohort Characterization and Groups Distribution

A total of 698 patients were included in this study. Three hundred and twenty patients (45.9%) had total thyroidectomy without lymph node resection (Group 1), 264 patients (37.8%) had total thyroidectomy with incidental lymph node resection without lymph node metastasis (Group 2), 37 patients (5.3%) had total thyroidectomy with incidental lymph node resection and lymph node metastasis (Group 3), 32 patients (4.6%) had a total thyroidectomy and CCND without lymph node metastasis (Group 4) and 45 patients (6.45%) had a total thyroidectomy and therapeutic CCND (Group 5). The data were analyzed according to each group distribution.

### 3.2. Clinical-Pathological Characteristics

Data regarding the age at diagnosis, sex, histologic subtype, tumor diameter, multifocality, microscopic extra-thyroidal extension, amongst other clinical-pathological characteristics and other prognostic factors such as AJCC staging and ATA 2015 risk of recurrence are shown in Table 1.

The patients from Groups 3 and 5 (with lymph node metastasis) were significantly younger with medians of 42 and 44 years; IQR: 17 and 28 years, respectively, against medians of 55, 52 and 50 years; IQR: 19, 18 and 23 years from the patients of Groups 1, 2 and 4 (*p* < 0.001; G1 vs. G3: *p* < 0.001; G2 vs. G3: *p* = 0.004; G1 vs. G5: *p* = 0.005—Kruskal–Wallis test, Dunn’s test amongst groups). In addition, they had larger tumor sizes with medians of 1.93 and 2.26 cm; IQR: 1.30 and 2.40 cm, respectively, against medians of 1.37, 1.35 and 1.93 cm; IQR: 1.12, 1.20 and 1.70 cm from the patients of Groups 1, 2 and 4 (*p* < 0.001; G1 vs. G3: *p* < 0.001; G2 vs. G3: *p* = 0.009; G1 vs. G4: *p* = 0.014; G1 vs. G5: *p* = 0.001—Kruskal–Wallis test, Dunn’s test amongst groups). Furthermore, other variables were also more frequent in these patients: microscopic extra-thyroidal extension (G3: 68%, G5: 64% against G1: 20%, G2: 28% and G4: 31%—*p* < 0.001—Chi-square test), angiolymphatic invasion (G3: 41%, G5: 36% against G1: 6.6%, G2: 6.8% and G4: 3.1%—*p* < 0.001—Chi-square test). Finally, they had higher pT classification (*p* < 0.001—Chi-square test), AJCC group staging (*p* < 0.001—Chi-square test) and more moderate or higher risk of recurrence (*p* < 0.001—Chi-square test).

There was no predominance of aggressive papillary carcinoma subtypes. Between the groups with incidental lymph node resection (Groups 2 and 3), the latter had a higher number of lymph nodes resected with medians of 1 and 2 lymph nodes; IQR: 1 and 2 lymph nodes, respectively (*p* < 0.001—Kruskal–Wallis test, Dunn’s test amongst groups), and there was no difference between this variable with the groups that had CCND (Groups 4 and 5). In addition, the incidental lymph node dissection rate was for the 264 patients of Group 2: 141 (53%), 70 (27%), 30 (11%) and 23 (9%) patients with, respectively 1, 2, 3 and ≥4 lymph nodes resected. For the 37 cases of Group 3: 7 (19%), 9 (24%), 10 (27%) and 11 (30%) patients with, respectively, 1, 2, 3 and ≥4 lymph nodes resected. The lymph node ratio was significantly higher in Group 3 (median 0.61 × 0.41 from Group 5, *p* = 0.002—Mann–Whitney U-test). The lymph node metastasis size median was 0.8 mm, varying from 0.2 to 11 mm in Group 3, and the frequency of extranodal extension was 10.8%. These variables were unavailable and not reviewed in Group 5; therefore, we considered 9 patients (20%) without other indicators of worsening recurrence risk stratification accordingly to ATA to be at least the moderate risk of recurrence in this group.

### 3.3. Treatment and Follow-Up Results

Most of the patients had at least one WBS after initial treatment or during the follow-up: two hundred and seven patients (64.7%) in Group 1, 183 patients (69.3%) in Group 2, 36 patients (97.3%) in Group 3, 25 patients (78.1%) in Group 4 and 43 patients (95.5%) on Group 5. WBS were predominantly performed during the radioiodine therapy period, and the groups did not differ in iodine uptake rates. The TSH had similar levels among the groups and stratified stimulated thyroglobulin >10 ng/mL were significantly more frequent in Groups 5 and 3 (23.3% and 16.7%, respectively, against 12.1%, 6% and 8% from Group 1, 2 and 4—*p* = 0.036—Chi-square test). The abnormal iodine uptake was present in all groups but was not statistically significant (*p* = 0.094—Chi-square test); it was more frequent in Group 5. Distant iodine uptake was present in Groups 1 and 5 and in the neck (central and lateral compartments) more frequently in Groups 3 and 5. Details of the first WBS are shown in Table 2.

The radioiodine therapy was performed in 156 patients (48.8%) of Group 1, 137 patients (51.9%) of Group 2, 36 patients (97.3%) of Group 3, 19 patients (59.4%) of Group 4 and 42 patients (93.3%) of Group 5. Most of the patients of Groups 1, 2 and 4 had significantly lower doses (remnant ablation) and Groups 3 and 5 had higher doses (adjuvant therapy), as described in Table 2. Medians: 150, 154 and 161; IQR: 54, 53 and 35 mCi, respectively, for Groups 1, 2 and 4 and medians 206 and 208; IQR: 15 and 13 mCi, respectively for Groups 3 and 5 (*p* < 0.001; G1, 2 vs. G3: *p* < 0.001; G1, 2, 4 vs. G5: *p* < 0.001; G3 vs. G4: *p* = 0.017—Kruskal–Wallis test, Dunn’s test amongst groups).

Thyroglobulin levels assessed on three occasions during the follow-up in each group (immediately after treatment, one year after initial treatment and in the last medical appointment) presented significantly different levels between groups and were slightly higher in Groups 3 and 5 (*p* = 0.016, 0.002, 0.002, respectively for each occasion—Kruskal–Wallis test—After treatment: G1 vs. G5 *p* = 0.027; After 1 year: G1 vs. G5 *p* = 0.002; G2 vs. G5 *p* < 0.001; G4 vs. G5 *p* = 0.007; Last appointment: G2 vs. G5 *p* = 0.005; G2 vs. G5 *p* = 0.001; G4 vs. G5 *p* = 0.012—Dunn’s test amongst groups). When stratified, thyroglobulin levels higher than 1 ng/mL occurred more often in Groups 3 and 5 one year after initial treatment and in the last medical appointment (*p* < 0.001 and *p* = 0.034, respectively—Chi-square test). The levels of anti-thyroglobulin antibodies were similarly present between all groups one year after initial treatment and in the last medical appointment (*p* = 0.728 and *p* = 0.259, respectively—Chi-square test).

There was no statistically significant difference between levels of thyroglobulin and anti-thyroglobulin before treatment failure identification amongst groups (*p* = 0.610—Kruskal–Wallis test and *p* = 0.322—Chi-square test, respectively). These results were stratified and are detailed in Table 3 and Figure 1.

Response-to-therapy categories were evaluated one year after initial treatment, and the patients of Group 5 had a significantly higher incidence of incomplete biochemical response (44%), followed by Group 3 (24%), which also had high rates of incomplete structural response (5.4%) (*p* < 0.001—Chi-square test). Both groups also significantly needed more radioiodine therapy sessions during follow-up (More than one radioiodine therapy session: 3.2%, 1.5%, 11%, 0%, 7.1%, respectively for Groups 1—5; *p* < 0.001—Chi-square test). Treatment failure (persistence or recurrence—local and/or locoregional or distant) was significantly higher in Groups 3 and 5 (32% and 16%, respectively, against 4.7%, 3% and 6.2% from Groups 1, 2 and 4; *p* < 0.001—Chi-square test), with Group 3 leading the locoregional recurrence in the central neck compartment. This group also needed significantly more reoperations than the other groups (27% against 6.6% from Group 5, followed by the other Groups; *p* < 0.001—Chi-square test. Distant metastases were higher in Group 5, but not statistically significant (8.9% against 5.4% in Group 3 and 2.2% in Group 1; *p* = 0.155—Chi-square test), and two patients (4.4%) needed radiotherapy and chemotherapy/target therapy as adjuvant treatment in this group. The total iodine-131 received dose was significantly higher in the patients from Groups 3 and 5 over the follow-up period (Medians 206 and 208; IQR: 16 and 15 mCi, respectively for Groups 3 and 5 and medians: 150, 154 and 161; IQR: 54, 54 and 35 mCi, respectively for Groups 1, 2 and 4—*p* < 0.001; G1, 2 vs. G3: *p* < 0.001; G1, 2 vs. G5: *p* < 0.001; G3 vs. G4: *p* = 0.011; G4 vs. G5: *p* = 0.005—Kruskal–Wallis test, Dunn’s test amongst groups).

One patient succumbed to a cancer-related death in Group 3, which was caused by perioperative complications after a CCND due to recurrence, and three patients died from thyroid cancer in Group 5. The follow-up time was similar amongst Groups 1 to 3 and 5, but longer for patients of Group 4 compared to Groups 1 and 2. Details of the follow-up characteristics and outcomes are shown in Table 4.

### 3.4. Disease-Free Survival

The main outcome evaluated was treatment failure, evaluating persistency or recurrence (local, locoregional and/or distant disease). The pairwise comparisons (Log-Rank test) showed no difference between the groups with no metastatic lymph node resection (Groups 1, 2 and 4). However, Group 3 had significantly lower DFS when compared to Groups 1, 2 and 4 (*p* < 0.001 vs. G1 and 2 and *p* = 0.005 vs. G4) but with similar DFS when compared to Group 5 (*p* = 0.091), as shown in Figure 2.

The Cox proportional-hazards regression model was used to identify variables related to lower DFS. The univariate analyses demonstrated that Groups 3 and 5, pT3a, pT3b, pT4a, microscopic extra-thyroidal extension and angiolymphatic invasion were associated with lower DFS, as described in Table 5. However, the multivariate analysis (Table 5) demonstrated as an independent risk factors for treatment failure the following variables: pT4a (HR = 5.524; 95%CI: 1.380–22.113; *p* = 0.016), Group 3 (HR = 3.691; 95%CI: 1.556–8.755; *p* = 0.003), microscopic extra-thyroidal extension (HR = 2.560; 95%CI: 1.303–5.030; *p* = 0.006) and angiolymphatic invasion (HR = 2.240; 95%CI: 1.077–4.510; *p* = 0.030). Variables such as nodal status (Group dependent), macroscopic extra-thyroidal extension (pT status dependent) and ATA risk of recurrence (multivariable dependent) were not included in the multivariate analysis.

Finally, higher rates of treatment failure identified in Group 3 have a discriminatory power of 99.1%, 99.6% and 78.3% in comparison to Groups 1, 2 and 4. However, for Group 5, this rate was only 40.1%.

### 3.5. Group 3 Assessment

According to the results demonstrating Group 3, in order words, patients with incidental metastases at thyroidectomy for PTC, as an independent risk factor for treatment failure, a thorough assessment of these patients was performed in an attempt to identify which of these patients are more likely to present this outcome. Regarding clinical and pathological characteristics, there were no statistically significant differences in terms of age at diagnosis, sex, tumor size, number of resected and metastatic lymph nodes, lymph node ratio, lymph node metastasis size, extranodal extension, PTC subtypes and risk of recurrence stratification groups according to ATA 2015 guidelines. However, the patients that presented treatment failure had higher pT status classification (pT status ≥ pT2: 41.6% vs. 28%; *p* = 0.0279—Chi-square test).

The number of WBS, its indications and iodine uptake rates were similar; stimulated thyroglobulin was higher in the treatment failure subgroup, and abnormal radioiodine uptake was present in the lateral cervical compartment in two patients only in the subgroup of patients without treatment failure. In both patients, it was identified in the radioiodine therapy period’s WBS, one had 13 months of follow-up and died of other causes and the other had no disease recurrence over 100 months of follow-up. Radioiodine therapy had similar indications and iodine-131 dose, mostly as adjuvant therapy. One patient (8.3%) had persistent disease identification before radioiodine therapy and the follow-up time was similar between the subgroups. Group 3’s exploratory characteristics stratified due to treatment failure are described in Table 6.

## 4. Discussion

Occasionally, after a total thyroidectomy, the pathological examination reveals incidentally resected lymph nodes eventually compromised by metastatic disease. This study presents not an uncommon scenario faced by many surgeons and endocrinologists in treating patients with PTC and demonstrates that patients with this characteristic present more treatment failure and have lower disease-free survival, despite radioiodine therapy with adjuvant intent in most cases. Furthermore, these patients needed more numbers of reoperations, radioiodine therapy sessions and consequently higher total doses. Finally, as expected, patients with lymph node metastasis during the follow-up also had a higher thyroglobulin curve. We consider these important findings aspects of the assessment and management of PTC patients.

Kluijfhout et al. [8] studied a cohort of 1000 patients with PTC in 2016 and 225 of those individuals who received a total thyroidectomy had incidentally resected lymph nodes. Forty-two of these patients had incidental lymph node metastasis and 16.7% had disease recurrence against 4.4% with no metastatic lymph nodes. The disease-free survival curves were significantly lower in the first group. The cumulative survival was 79% (CI 95%: 58–90%) for patients with incidental lymph node metastasis versus 96% (CI 95%: 89–98%) with incidental lymph resection but no metastasis in 60 months. The incidental lymph node metastasis group was also independently associated with disease recurrence in the multivariate analyses (HR = 5.19; CI 95%: 1.74–15.5). These results are similar to ours in terms of DFS and risk association. Moreover, the authors found that 69.1% of the patients with incidental lymph node metastasis had radioiodine therapy. Otherwise, in our cohort, almost all the patients from Group 3 received it, most as adjuvant treatment. Moreover, in the study of Kluijfhout et al. [8], patients with incidental lymph node metastasis also had significantly more extra-thyroidal extension, results again similar to ours, but with similar angiolymphatic invasion.

Considering the role of radioiodine therapy in patients with incidental lymph node metastasis, in one of the few studies available, Wang et al. [9], in 2016, in a cohort of 248 patients, after exclusion of individuals with other indications of radioiodine adjuvant therapy (i.e., extra-thyroidal extension, close or positive margins) evaluated 104 patients wherein 67 underwent adjuvant radioiodine therapy (mostly due to angiolymphatic invasion, metastasis > 2 mm and more than two compromised lymph nodes). They demonstrated that there was no difference in terms of cumulative disease-free survival (94.6% vs. 96.2%) over 60 months. Their finding contrasts with ours in which the comparable group (Group 3) had 71% cumulative survival in the same period. Despite our smaller population, this might be explained by the fact that their cohort was mostly composed of patients with early tumors (pT1-2) and the exclusion of patients with extra-thyroidal extension, differently from the patients of our study.

The findings of both previously discussed studies led us to an impasse. The first [8] was concordant with our findings, and the other [9] was not. Although our results indicate that Group 3 is independently associated with lower disease-free survival, our critical view of this finding is that these patients are in a gap of a dichotomous spectrum in the disease’s natural history. On one side of this spectrum, there is an earlier disease in which the tumor is smaller, unifocal or has few foci, with or without a few angiolymphatic invasions and extra-thyroidal extension spots. On the other side of the spectrum, the opposite, where the metastatic lymph node disease is higher in number and volume. In general, not considering the genetic characteristics and the tumor microenvironment, which must also play a key role in lymph node metastases [10,11,12,13,14,15,16] and are not addressed in this study, the likely and logical pathway for developing more metastatic lymph node disease are that the higher the tumor volumes, foci, angiolymphatic invasions and extra-thyroidal extensions, the higher the probability of lymph node involvement. This is already documented in the literature [3,4,17,18,19]. There is also evidence that microscopic, extra-thyroidal extension, which was down-staged from the TNM8 T3 tumor classification, affects recurrence and response-to-therapy categories but does not affect overall survival [20].

The rationale of the previous point of view suggests that the patients from Group 3 probably had more occult lymph node metastasis in the central compartment with a critical impact on disease-free survival, as most of the recurrences occurred in this compartment. Although it is not evident nor feasible to conclude from our study, we hypothesize that these occult lymph node metastases could also impact disease-specific survival in the long term. Shi et al. [21], in 2018, demonstrated that the possible presence of occult lymph node metastasis in the central compartment led to worse overall survival and cancer-specific survival in patients with PTC tall-cell variant subtype in terms of the number of examined lymph nodes. The patients clinically N0 without lymph nodes on the pathological examination had similar outcomes. However, patients pN1a and pN0 with two or more examined lymph nodes had better outcomes than both, even in the initial stages (T1–2) and after adjuvant radioiodine therapy. This study focused on a specific and aggressive PTC variant, but the concept might be applicable.

The high rates of metastatic lymph nodes in the central compartment provided a wide field for the creation of various parameters in order to stratify pN1a patients in terms of recurrence, such as metastatic lymph node size, foci size and extra-nodal extension [3,6,22]. Another proposed parameter to estimate the possibility of occult lymph node metastasis in the central compartment is the lymph node ratio, which is the number of metastatic lymph nodes divided by the number of total resected lymph nodes. Pyo et al. [23], in 2018, in a retrospective study and meta-analysis, demonstrated that a ≥0.44 lymph node ratio was significantly associated with macro metastases (>2 mm) and extra-nodal extension and correlated it with worse disease-free survival. 

In our study, the patients from Group 3 had lymph node ratios slightly higher than this cutoff. However, one must observe that most of the studies regarding the lymph node ratio came from the realization of central compartment neck dissections (elective and therapeutic) where more lymph nodes are removed or indiscriminating incidental lymph node resection from the systematical neck dissection. Therefore, we cannot assume this number to be interpreted alone, especially in the incidental lymph node resection scenario, as most of these patients had 2 to 3 lymph nodes removed. Interestingly, in the topic of the lymph node yield necessary to lower the chance of having occult lymph node metastasis, Robinson et al. [24], in 2016, in a large population-based study, sought to stratify the number of necessary examined lymph nodes according to the pT classification and proposed that 3, 4 and 8 lymph nodes in the central compartment are needed to be examined to rule out, respectively, occult lymph node metastasis for pT1b, pT2 and pT3 patients.

It is well established that patients with PTC with evident metastatic lymph nodes in the central compartment should undergo therapeutic neck dissection. However, the role of this procedure in patients with clinically negative lymph nodes remains uncertain. We do not routinely perform elective central compartment neck dissection at our institution. The patients from Groups 4 and 5 that underwent this procedure were part of previous clinical trials [25,26] or mostly indicated that in terms of preoperative suspicious lymph nodes or other intraoperative findings, such as a gross extra-thyroidal extension or clinically suspicious lymph nodes, no pathologist was available to perform the frozen section at the time of the surgery. Hence, the ATA 2015 guidelines [3] have a weak recommendation with low-quality evidence to support elective central compartment neck dissection in patients with more advanced tumors (pT3-4) and involved lateral neck lymph nodes. After the release of these guidelines, several novel studies either supported or discouraged the procedure. Most of these studies clearly demonstrate the decrease in locoregional recurrence when the central compartment dissection was performed. Still, there is no evidence of the impact on overall survival or disease-specific survival. The studies that discouraged the procedure [27,28,29,30,31,32,33,34] emphasized that the morbidities (especially hypoparathyroidism and recurrent laryngeal nerve injury) overcome this benefit and pointed out that the only benefit was to upstage the disease and provide information for adequate radioiodine therapy dosage. The others that encouraged the procedure [35,36,37] defended its benefits, especially for patients with higher-risk tumors (larger, with angiolymphatic invasion and/or extra-thyroidal extent).

In our study, the participants in Group 3 already had lymph node specimens in the pathological examination that upstaged the node disease status, providing enough information for adequate radioiodine therapy dosage when indicated. Thus, we also questioned the possibility or necessity of submitting these patients to reoperation to perform a therapeutic CCND. In terms of this topic, Hall et al. [37], evaluating a retrospective cohort of 283 patients, verified that the patients submitted to CCND reoperation (elective or therapeutic) had similar morbidities in terms of permanent hypoparathyroidism and recurrent laryngeal nerve injury compared to those submitted to this procedure in the first surgery. The indications of elective CCND were factors such as age > 45 years old, tumors large than 1 cm, extra-thyroidal extension and angiolymphatic invasion. The overall recurrence in this study was 2% (follow-up 61 ± 29.7 months), without recurrence in the group which underwent elective CCND and 11% in those that had a therapeutic procedure. The results in terms of recurrence are similar to our comparable group (Group 5).

Although there is evidence of the potential lower recurrence rate in the possibility of an interval therapeutic CCND (a programmed therapeutic reoperation after an incidental metastatic lymph node resection) since it is not clear which of the patients with incidental lymph node metastasis (Group 3) are more likely to present treatment failure we consider that the indication of this procedure could set these patients to potential morbidity before they really need it. Nevertheless, as data are not sufficient to determine the impact on overall survival (only in disease-specific survival), we believe that this decision should be tailored to the patient by the endocrinologist and the surgeon in a shared decision-making process considering the risk of recurrence, patient’s desires and perspectives, socioeconomic aspect, and access to the health care system as they will need a close follow-up with physical examination, neck ultrasound and thyroglobulin tests to detect early signs of disease recurrence.

### Limitations

An important limitation of our study, even with a five-year cohort in a high-volume medical center, is the small number of patients in Groups 3 to 5. In addition, the only statistically significant factor more frequent in patients with treatment failure from Group 3 was the pT status. We believe that other variables, such as lymph node metastasis size, extranodal and extra-thyroidal extension, could be statistically significant with a larger sample. This demonstrates the necessity of further, probably multicentric, collaborative studies to validate these findings.

The reader may question that the incidental lymph node detection and rare or more aggressive PTC subtypes detection are possibly lower compared to other studies in this cohort. However, we utilized the pathological report and were not able to perform further pathological reviews to attest to this, and this limitation favors us in dealing with a real-life clinical practice scenario where the surgeons and endocrinologists must trust the pathological report to manage their patients.

A considerable number of patients, especially from Groups 1 and 2, but also from other groups, had surgery for other indications aside from suspected or confirmed malignant tumors (e.g., nodular goiter with compressive symptoms), which explains the high incidence of incidentally diagnosed microcarcinoma in these groups. Some of them were probably of indolent behavior, but some also had intraoperative findings of suspected lymph node metastasis, as occurred in one case of Group 5. Therefore, we emphasize the comparisons between Groups 3 and 5.

Despite the high rate of radioiodine therapy with adjuvant doses in the patients of Group 3, this study cannot conclude that this is a protector factor, nor else discard its utility since it is unclear if more patients would present treatment failure with lower iodine-131 doses. The reader might notice the use of this therapy in cases that nowadays would not be indicated in this cohort. Its indications were more liberal in the past. Moreover, our institutional protocols have changed over time as novel studies and guidelines have been published. However, we must consider that the surgeon and endocrinologist’s individual decision-making experience might also have been a determinant of the indication of adjuvant radioiodine therapy in these cases.

Moreover, the tumor genetic profile and its relationship with incidental lymph node metastasis were not addressed in this study, warranting further studies on behalf of this aspect.

Finally, the proper retrospective nature of this study possibly underestimates the DFS as the date of treatment failure cannot be precisely determined. Based on that, these intervals should be interpreted with caution. Moreover, the rareness of the incidental lymph node metastasis finding exclusively in the pathological examination and long follow-up periods could complicate the viability of prospective studies in this field.

## 5. Conclusions

Incidental lymph node metastasis identification exclusively in the pathological examination after total thyroidectomy in patients with PTC was independently associated with lower disease-free survival, despite radioiodine therapy with adjuvant doses in most cases. These patients are probably in the gap between low-volume lymph node disease, with lower impact in DFS, and high-volume lymph node disease, which was not detected by imaging or intraoperatively and was not properly treated during the surgery.

To the present, data is insufficient to consider these patients to be submitted to reoperation and perform a formal CCND after initial surgery before persistent or recurrent disease identification. Moreover, there is no compelling evidence of impact on overall survival, which would require a huge cohort of patients. This procedure could lead to more perioperative or definitive complications. We encourage a systematic central compartment evaluation by the surgeon and, when feasible, the peri-isthmus fat tissue by the pathologist during the surgery for additional information on lymph node metastasis, and an extrathyroidal extension could possibly change intraoperative management. Foremost, we believe these patients must be closely followed with periodical physical examinations, neck ultrasounds and monitoring of thyroglobulin levels to detect signs of disease recurrence as early as possible to provide effective and prompt treatment.

## Figures and Tables

**Figure 1 cancers-15-00943-f001:**
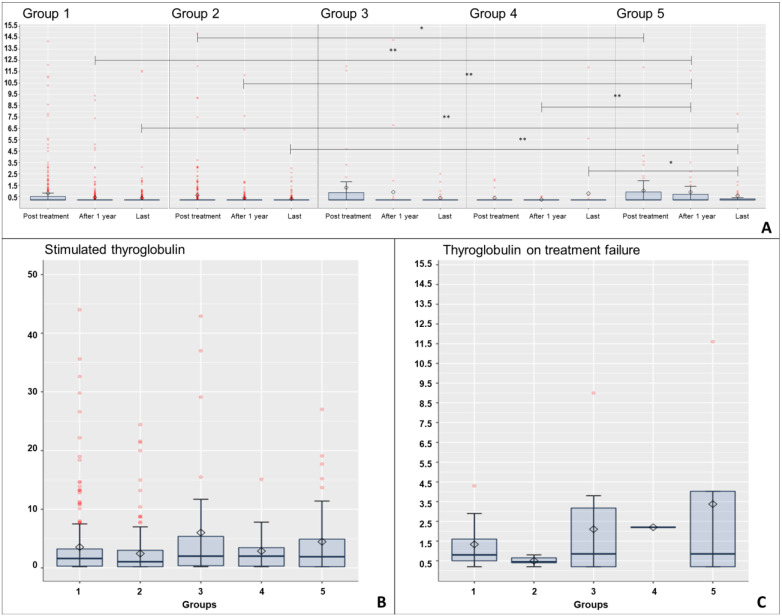
Boxplot charts demonstrating thyroglobulin levels (ng/mL) amongst groups. (**A**) Thyroglobulin levels on three occasions during the follow-up in each group: immediately after treatment (surgery only or after radioiodine therapy), one year after initial treatment and in the last medical appointment demonstrating significantly different levels between groups as indicated by comparative lines. (**B**) Stimulated thyroglobulin levels during the first whole-body scintigraphy examination presenting similar levels (*p* = 0.087—Kruskal–Wallis test). (**C**) Thyroglobulin levels assessed immediately before treatment failure identification presenting similar levels (*p* = 0.610—Kruskal–Wallis test). Note: Group 1: Total thyroidectomy without incidental lymph node resection; Group 2: Total thyroidectomy with incidental lymph node resection without lymph node metastasis; Group 3: Total thyroidectomy with incidental lymph node resection and lymph node metastasis; Group 4: Total thyroidectomy with central compartment neck dissection without lymph node metastasis; Group 5: Total thyroidectomy with therapeutic central compartment neck dissection (with lymph node metastasis); * *p* < 0.05 and ** *p* < 0.01 (Kruskal–Wallis test—Dunn’s test amongst groups).

**Figure 2 cancers-15-00943-f002:**
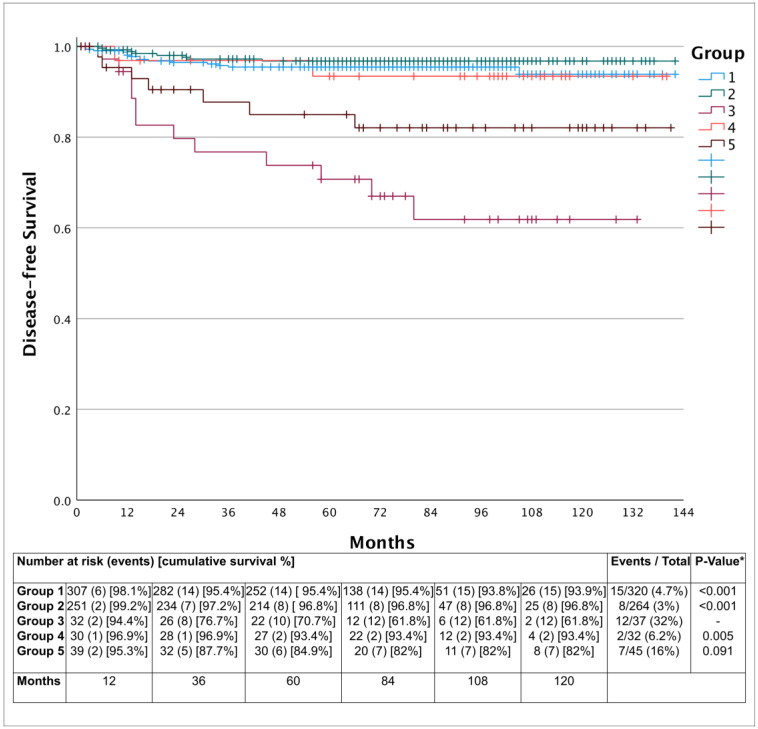
Kaplan–Meier’s curves demonstrating lower disease-free survival in the patients of Groups 3 and 5. Cumulative disease-free survival of 93.9% (15 events in 320 cases), 96.8% (8 events in 264 cases), 61.8% (12 events in 37 cases), 93.4% (2 events in 32 cases) and 82% (7 events in 45 cases), respectively for groups 1 to 5 (Group 3, *p* < 0.001 vs. Groups 1 and 2, *p* = 0.005 vs. Group 4 and *p* = 0.091 vs. Group 5—* Log-Rank test). Note: Group 1: Total thyroidectomy without incidental lymph node resection; Group 2: Total thyroidectomy with incidental lymph node resection without lymph node metastasis; Group 3: Total thyroidectomy with incidental lymph node resection and lymph node metastasis; Group 4: Total thyroidectomy with central compartment neck dissection without lymph node metastasis; Group 5: Total thyroidectomy with therapeutic central compartment neck dissection (with lymph node metastasis).

**Table 1 cancers-15-00943-t001:** Clinical-pathological characteristics of the patients included in the cohort.

	Group 1: 320 (45.8%) ^1^	Group 2: 264 (37.8%) ^1^	Group 3: 37 (5.3%) ^1^	Group 4: 32 (4.5%) ^1^	Group 5: 45 (6.4%) ^1^	*p*-Value
Age						
Mean (SD)	53 (13)	51 (14)	42 (12)	50 (17)	45 (17)	<0.001 ** Group 1 vs. 3: <0.001 Group 2 vs. 3: 0.004 Group 1 vs. 5: 0.005
Sex						
Male	39 (12%)	22 (8.3%)	7 (19%)	4 (12%)	5 (11%)	0.307 ***
Female	281 (88%)	242 (92%)	30 (81%)	28 (88%)	40 (89%)	
CCND Indication						
Therapeutic (preoperative)	-	-	-	-	3 (6.7%)	
Therapeutic (intraoperative)	-	-	-	-	28 (62%)	
Elective (prophylactic)	-	-	-	32 (100%)	14 (31%)	-
Tumor size-largest foci (cm)						
Mean (SD)	1.37 (1.51)	1.35 (1.20)	1.93 (1.35)	1.93 (1.50)	2.26 (1.93)	<0.001 ** Group 1 vs. 3: <0.001 Group 2 vs. 3: 0.009 Group 1 vs. 4: 0.014 Group 1 vs. 5: <0.001
Median (IQR)	0.90 (0.40, 1.52)	1.00 (0.50, 1.70)	1.50 (1.10, 2.40)	1.50 (0.80, 2.50)	1.50 (1.00, 3.40)	
Range (min, max)	0.03, 9.00	0.01, 8.30	0.30, 6.40	0.50, 7.00	0.01, 10.50	
Tumor multifocality ****	145 (45%)	128 (48%)	26 (70%)	16 (50%)	23 (51%)	0.075 ***
Microscopic ETE	63 (20%)	73 (28%)	25 (68%)	10 (31%)	29 (64%)	<0.001 ***
Angiolymphatic invasion	21 (6.6%)	18 (6.8%)	15 (41%)	1 (3.1%)	16 (36%)	<0.001 ***
pT status AJCC 8th ed.						
pT1a	184 (57%)	137 (52%)	8 (22%)	11 (34%)	12 (27%)	<0.001 ***
pT1b	71 (22%)	73 (28%)	17 (46%)	9 (28%)	17 (38%)	
pT2	43 (13%)	35 (13%)	7 (19%)	7 (22%)	5 (11%)	
pT3a	17 (5.3%)	12 (4.5%)	1 (2.7%)	4 (12%)	1 (2.2%)	
pT3b	4 (1.2%)	7 (2.7%)	3 (8.1%)	1 (3.1%)	5 (11%)	
pT4a	1 (0.3%)	-	1 (2.7%)	-	5 (11%)	
Group staging AJCC 8th ed.						
I	312 (98%)	256 (97%)	33 (89%)	30 (94%)	33 (73%)	<0.001 ***
II	7 (2.2%)	8 (3.0%)	3 (8.1%)	2 (6.2%)	9 (20%)	
III	-	-	1 (2.7%)	-	3 (6.7%)	
IVb	1 (0.3%)	-	-	-	-	
Resected nodes						
Mean (SD)	-	2 (1)	3 (2)	8 (5)	10 (5)	<0.001 ** Group 2 vs. 3: <0.001 Group 2 vs. 4: <0.001 Group 3 vs. 4: <0.001 Group 2 vs. 5: <0.001 Group 3 vs. 5: <0.001
Median (IQR)	-	1 (1, 2)	3 (2, 4)	7 (5, 10)	9 (6, 14)	
Range (min; max)	-	1; 9	1; 8	3; 22	2; 24	
1 node	-	141 (53%)	7 (19%)	-	-	<0.001 ***
2 nodes	-	70 (27%)	9 (24%)	-	1 (2.2%)	
3 nodes	-	30 (11%)	10 (27%)	3 (9.4%)	3 (6.7%)	
≥4 nodes	-	23 (9%)	11 (30%)	29 (90.6%)	41 (91.1%)	
Metastatic nodes						
Mean (SD)	-	-	1 (1)	-	4 (3)	<0.001 **
Median (IQR)	-	-	1 (1, 2)	-	3 (2, 4)	
Range (min; max)	-	-	1; 4	-	1; 18	
1 node	-	-	24 (65%)	-	11 (24%)	<0.001 ***
2 nodes	-	-	10 (27%)	-	10 (22%)	
3 nodes	-	-	2 (5.4%)	-	7 (16%)	
≥4 nodes	-	-	1 (2.7%)	-	17 (38%)	
Lymph node ratio						
Mean (SD)	-	-	0.61 (0.31)	-	0.41 (0.27)	0.002 **
Node metastasis size (mm)						
Mean (SD)	-	-	2.08 (2.46)	-	NA	-
Median (IQR)	-	-	0.80 (0.50, 3)	-	NA	
Range (min, max)	-	-	0.20, 11	-	NA	
Extranodal extension	-	-	4 (10.8%)	-	NA	-
PTC subtype						
Classic	151 (47%)	151 (57%)	29 (78%)	19 (59%)	34 (76%)	<0.001 ***
Follicular	155 (48%)	100 (38%)	5 (14%)	10 (31%)	8 (18%)	
Oncocytic	8 (2.5%)	7 (2.7%)	1 (2.7%)	3 (9.4%)	1 (2.2%)	
Solid forms	4 (1.2%)	1 (0.4%)	1 (2.7%)	-	-	
Tall cell	1 (0.3%)	3 (1.1%)	-	-	-	
Columnar cells	1 (0.3%)	-	-	-	-	
Diffuse sclerosing	-	-	-	-	2 (4.4%)	
Other	-	2 (0.8%)	1 (2.7%)	-	-	
Risk of recurrence *****						
Low risk	238 (74%)	186 (70%)	4 (11%)	21 (66%)	-*	<0.001 ***
Moderate risk	76 (24%)	73 (28%)	31 (84%)	10 (31%)	3 (78%)	
High risk	6 (1.9%)	5 (1.9%)	2 (5.4%)	1 (3.1%)	10 (22%)	

Note: Group 1: Total thyroidectomy without lymph node resection; Group 2: Total thyroidectomy with incidental lymph node resection but no lymph node metastasis; Group 3: Total thyroidectomy with incidental lymph node resection with lymph node metastasis; Group 4: Total thyroidectomy with central compartment neck dissection (elective) but no lymph node metastasis; Group 5: Total thyroidectomy with central compartment neck dissection (elective or therapeutic) with lymph node metastasis. Legend: ^1^ n (%); AJCC—American Joint Committee on Cancer; CCND—Central compartment neck dissection; WBS—Whole-body scintigraphy; RDIT—Radioiodine therapy; ETE—Extra thyroidal extension; PTC—papillary thyroid carcinoma; SD—standard deviation; SD—Standard deviation; IQR—Inter-quantile range; NA—Not available; * Since the LN metastasis specimens were not reviewed in Group 5 and nodes metastasis size and extra-nodal extension information were not available; we considered 9 patients (20%) without other indicators of worsening risk stratification to be at least as a moderate risk of recurrence; ** Kruskal–Wallis test (Dunn’s test amongst groups); *** Chi-square test; **** Regarding the multiples tumor foci which may present within the thyroid gland as a result of either intrathyroidal tumor metastasis or coexistence of separate neoplastic foci; ***** According to ATA: American Thyroid Association 2015 guidelines.

**Table 2 cancers-15-00943-t002:** Initial whole-body scintigraphy and initial radioiodine therapy characteristics.

	Group 1: 320 (45.8%) ^1^	Group 2: 264 (37.8%) ^1^	Group 3: 37 (5.3%) ^1^	Group 4: 32 (4.5%) ^1^	Group 5: 45 (6.4%) ^1^	*p*-Value
WBS	207 (64.7%)	183 (69.3%)	36 (97.3%)	25 (78.1%)	43 (95.5%)	<0.001 **
Indication						
Diagnostic WBS	58 (28%)	55 (30%)	1 (2.9%)	7 (28%)	-	<0.001 **
RDIT period	146 (72%)	127 (70%)	33 (97.1%)	18 (72%)	42 (100%)	
Abnormal iodine capitation	15 (7.3%)	6 (3.4%)	2 (5.7%)	1 (4.0%)	6 (14%)	
Cervical central compartment	3 (20%)	3 (50%)	-	-	1 (16.7%)	0.094 **
Cervical lateral compartment	3 (20%)	3 (50%)	2 (100%)	1 (100%)	3 (50%)	
Distant	9 (60%)	-	-	-	3 (50%)	
RDIT	156 (48.8%)	137 (51.9%)	36 (97.3%)	19 (59.4%)	42 (93.3%)	<0.001 **
Indication						
Remnant ablation	141 (90%)	127 (93%)	7 (19%)	17 (94%)	11 (26%)	<0.001 **
Adjuvant therapy	11 (7.1%)	10 (7.3%)	29 (81%)	1 (5.6%)	31 (74%)	
Therapy of persistent disease	4 (2.6%)	-	-	-	-	
Dose ^2^						
Mean (SD)	142 (44)	144 (33)	202 (28)	155 (33)	206 (24)	<0.001 * Group 1 vs. 3: <0.001 Group 2 vs. 3: <0.001 Group 3 vs. 4: 0.017 Group 1 vs. 5: <0.001 Group 2 vs. 5: <0.001 Group 4 vs. 5: <0.001
Median (IQR)	150 (105, 159)	154 (107, 160)	206 (200, 213)	161 (132, 167)	208 (200, 215)	
Range (min, max)	100, 382	30, 230	104, 266	100, 217	150, 272	

Note: Group 1: Total thyroidectomy without lymph node resection; Group 2: Total thyroidectomy with incidental lymph node resection but no lymph node metastasis; Group 3: Total thyroidectomy with incidental lymph node resection with lymph node metastasis; Group 4: Total thyroidectomy with central compartment neck dissection (elective) but no lymph node metastasis; Group 5: Total thyroidectomy with central compartment neck dissection (elective or therapeutic) with lymph node metastasis. Legend: ^1^ n (%); ^2^ mCi (millicuries); WBS—Whole-body scintigraphy; RDIT—Radioiodine therapy; SD—Standard deviation; IQR—Inter-quantile range; * Kruskal–Wallis test (Dunn’s test amongst groups); ** Chi-square test.

**Table 3 cancers-15-00943-t003:** Comparison of thyroglobulin and anti-thyroglobulin antibody levels between groups under adequate thyroid stimulation hormone suppression.

	Group 1: 320 (45.8%) ^1^	Group 2: 264 (37.8%) ^1^	Group 3: 37 (5.3%) ^1^	Group 4: 32 (4.5%) ^1^	Group 5: 45 (6.4%) ^1^	Total: 698 (100%) ^1^	*p*-Value *
Thyroglobulin after 1 year							
≤0.2 ng/mL	253 (79.1%)	209 (79.2%)	28 (75.7%)	26 (81.3%)	24 (53.3%)	540 (77.4%)	<0.001
>0.2−1 ng/mL	38 (11.9%)	33 (12.5%)	2 (5.4%)	3 (9.4%)	7 (15.5%)	83 (11.9%)	
>1−10 ng/mL	16 (2.8%)	8 (3%)	2 (5.4%)	-	7 (15.5%)	33 (4.7%)	
>10 ng/mL	1 (0.3%)	1 (0.4%)	2 (5.4%)	1 (3.1%)	2 (4.4%)	7 (1%)	
Missing	12 (3.8%)	13 (4.9%)	3 (8.1%)	2 (6.3%)	5 (11.1%)	35 (5%)	
Treatment failure	15 (4.6%)	8 (3%)	12 (32.4%)	2 (6.2%)	7 (15.5%)	44 (6.3%)	<0.001
Thyroglobulin before failure							
≤0.2 ng/mL	2 (13.3%)	1 (12.5%)	5 (41.6%)	-	2 (28.5%)	10 (22.7%)	0.060
>0.2−1 ng/mL	4 (26.6%)	5 (62.5%)	-	-	-	9 (20.4%)	
>1−10 ng/mL	3 (20%)	-	5 (41.6%)	1 (50%)	1 (14.3%)	10 (22.7%)	
>10 ng/mL	-	2 (25%)	1 (8.3%)	1 (50%)	1 (14.3%)	5 (11.4%)	
Missing	6 (40%)	-	1 (8.3%)	-	3 (42.9%)	10 (22.7%)	
Last thyroglobulin							
≤0.2 ng/mL	260 (81.3%)	217 (82.2%)	28 (75.7%)	27 (84.4%)	25 (55.6%)	557 (79.8%)	0.034
>0.2−1 ng/mL	41 (12.8%)	30 (11.4%)	4 (10.8%)	2 (6.3%)	10 (22.2%)	87 (12.5%)	
>1−10 ng/mL	9 (2.8%)	8 (3%)	2 (5.4%)	1 (3.1%)	3 (6.6%)	23 (3.3%)	
>10 ng/mL	5 (1.6%)	1 (0.4%)	1 (2.7%)	1 (3.1%)	3 (6.7%)	11 (1.6%)	
Missing	5 (1.6%)	8 (3%)	2 (5.4%)	1 (3.1%)	4 (8.9%)	20 (2.9%)	
Initial Anti-thyroglobulin							
Negative	273 (85.3%)	224 (84.8%)	29 (78.4%)	29 (90.6%)	38 (84.4%)	593 (85%)	0.728
Positive	41 (12.8%)	39 (14.8%)	7 (18.9%)	3 (9.4%)	5 (11.1%)	95 (13.6%)	
Missing	6 (1.9%)	1 (0.4%)	1 (2.7%)	-	2 (4.4%)	10 (1.4%)	
Anti-thyroglobulin before failure							
Negative	9 (2.8%)	5 (1.9%)	9 (24.3%)	2 (6.3%)	3 (6.7%)	28 (4%)	0.322
Positive	-	3 (1.1%)	2 (5.4%)	-	1 (2.2%)	6 (0.9%)	
Missing	6 (1.9%)	-	1 (2.7%)	-	3 (6.7%)	10 (1.4%)	
Last Anti-thyroglobulin							
Negative	305 (95.3%)	242 (91.7%)	32 (86,5%)	29 (90.6%)	40 (88.9%)	648 (92.8%)	0.259
Positive	10 (3.1%)	14 (5.3%)	3 (8.1%)	3 (9.4%)	1 (2.2%)	31 (4.4%)	
Missing	5 (1.6%)	8 (3%)	2 (5.4%)	-	4 (8.9%)	19 (2.7%)	
WBS	207 (64.7%)	183 (69.3%)	36 (97.3%)	25 (78.1%)	43 (95.6%)	494 (70.8%)	<0.001
Stimulated thyroglobulin							
<1 ng/mL	82 (39.6%)	87 (47.5%)	13 (36.1%)	8 (32%)	15 (34.9%)	205 (41.5%)	0.036
1−10 ng/mL	99 (47.8%)	80 (43.7%)	16 (44.4%)	15 (60%)	17 (39.5%)	227 (46%)	
>10 ng/mL	25 (12.1%)	11 (6%)	6 (16.7%)	2 (8%)	10 (23.3%)	54 (10.9%)	
Missing	1 (0.5%)	5 (2.7%)	1 (2.8%)	-	1 (2.3%)	8 (1.6%)	

Note: Group 1: Total thyroidectomy without lymph node resection; Group 2: Total thyroidectomy with incidental lymph node resection but no lymph node metastasis; Group 3: Total thyroidectomy with incidental lymph node resection with lymph node metastasis; Group 4: Total thyroidectomy with central compartment neck dissection (elective) but no lymph node metastasis; Group 5: Total thyroidectomy with central compartment neck dissection (elective or therapeutic) with lymph node metastasis. Legend: ^1^ n (%); WBS—Whole-body scintigraphy; * Chi-square test.

**Table 4 cancers-15-00943-t004:** Follow-up characteristics and outcomes.

	Group 1: 320 (45.8%) ^1^	Group 2: 264 (37.8%) ^1^	Group 3: 37 (5.3%) ^1^	Group 4: 32 (4.5%) ^1^	Group 5: 45 (6.4%) ^1^	*p*-Value
WBS sessions						
1	180 (87%)	169 (92%)	25 (69%)	24 (96%)	30 (70%)	<0.001 *
2	24 (12%)	14 (7.7%)	8 (22%)	1 (4.0%)	12 (28%)	
3	2 (1.0%)	-	2 (5.6%)	-	1 (2.3%)	
5	1 (0.5%)	-	1 (2.8%)	-	-	
RDIT sessions						
1	151 (97%)	135 (99%)	32 (89%)	19 (100%)	39 (93%)	0.041 *
2	5 (3.2%)	2 (1.5%)	4 (11%)	-	3 (7.1%)	
Treatment response ***						
Excellent response	237 (74%)	197 (75%)	20 (54%)	25 (78%)	20 (44%)	<0.001 *
Biochemical incomplete	62 (19%)	40 (15%)	9 (24%)	2 (6.2%)	20 (44%)	
Structural incomplete	1 (0.3%)	1 (0.4%)	2 (5.4%)	1 (3.1%)	1 (2.2%)	
Indeterminate	20 (6.2%)	26 (9.8%)	6 (16%)	4 (12%)	4 (8.9%)	
RDIT total dose ^2^						
Mean (SD)	149 (66)	147 (41)	225 (74)	155 (33)	222 (67)	<0.001 ** Group 1 vs. 3: <0.001 Group 2 vs. 3: <0.001 Group 3 vs. 4: 0.011 Group 1 vs. 5: <0.001 Group 2 vs. 5: <0.001 Group 4 vs. 5: 0.005
Median (IQR)	150 (105, 159)	154 (107, 161)	206 (202, 218)	161 (132, 167)	208 (200, 215)	
Range (min, max)	100, 590	30, 358	104, 472	100, 217	150, 502	
Treatment failure	15 (4.7%)	8 (3.0%)	12 (32%)	2 (6.2%)	7 (16%)	<0.001 *
Site						
Central compartment	6 (40%)	4 (50%)	9 (75%)	1 (50%)	1 (14.3%)	0.155 *
Lateral compartment	4 (26.6%)	4 (50%)	5 (41.6%)	1 (50%)	5 (71.4%)	
Distant metastasis	7 (2.2%)	-	2 (5.4%)	-	4 (8.9%)	
Reoperations	8 (2.5%)	9 (3.4%)	10 (27%)	1 (3.1%)	3 (6.6%)	<0.001 *
Radiotherapy	-	-	-	-	2 (4.4%)	-
Chemo/target therapy	-	-	-	-	2 (4.4%)	-
Death	16 (5.0%)	15 (5.7%)	2 (5.4%)	3 (9.4%)	3 (6.7%)	0.880 *
Cause of death						
Thyroid cancer	-	-	1 (50%)	-	3 (100%)	<0.001 *
Other	16 (100%)	14 (100%)	1 (50%)	3 (100%)	-	
Follow-up (months)						
Mean (SD)	81 (29)	80 (31)	82 (31)	96 (32)	83 (39)	<0.001 ** Group 1 vs. 4: 0.013 Group 2 vs. 4: 0.014
Median (IQR)	81 (66, 98)	80 (66, 103)	88 (69, 105)	102 (88, 115)	88 (67, 117)	
Range (min, max)	1, 142	0, 142	2, 133	10, 140	1, 141	

Note: Group 1: Total thyroidectomy without incidental lymph node resection; Group 2: Total thyroidectomy with incidental lymph node resection without lymph node metastasis; Group 3: Total thyroidectomy with incidental lymph node resection and lymph node metastasis; Group 4: Total thyroidectomy with central compartment neck dissection without lymph node metastasis; Group 5: Total thyroidectomy with therapeutic central compartment neck dissection (with lymph node metastasis). Legend: ^1^ n (%); ^2^ mCi (millicuries); WBS: Whole-body scintigraphy; RDIT—Radioiodine therapy; SD—Standard deviation; IQR—Inter-quantile range; * Chi-square test; ** Kruskal–Wallis test (Dunn’s test amongst groups); *** Evaluated one year after initial treatment according to American Thyroid Association’s 2015 guidelines response-to-therapy categories.

**Table 5 cancers-15-00943-t005:** Variables associated with worse disease-free survival in the univariate and multivariate analyses.

VARIABLES	UNIVARIATE			MULTIVARIATE		
	HR ^1^	CI ^2^ 95%	*p*-Value ^3^	HR ^1^	CI ^2^ 95%	*p*-Value ^3^
Sex: Male	0.607	0.270–1.364	0.227	-	-	-
Age (≥55 years)	0.918	0.501–1.683	0.782	-	-	-
Group 1	Ref.	-	-	Ref.	-	-
Group 2	0.692	0.290–1.649	0.406	0.579	0.245–1.317	0.214
Group 3	8.476	3.918–18.336	<0.001	3.691	1.556–8.755	0.003
Group 4	1.378	0.313–6.068	0.672	1.141	0.259–5.026	0.862
Group 5	3.923	1.583–9.725	0.003	1.749	0.649–4.709	0.269
pT1a ^4^	Ref.	-	-	Ref.	-	-
pT1b ^4^	1.891	0.877–4.079	0.104	0.934	0.395–2.207	0.876
pT2 ^4^	1.727	0.656–4.544	0.268	0.997	0.352–2.825	0.995
pT3a ^4^	2.980	0.971–9.140	0.056	1.981	0.588–6.082	0.285
pT3b ^4^	4.282	1.220–15.032	0.023	1.947	0.570–6.653	0.288
pT4a ^4^	21.865	7.119–67.154	<0.001	5.524	1.380–22.113	0.016
Multifocality	1.335	0.732–2.438	0.346	-	-	-
Microscopic ETE	3.990	2.165–7.353	<0.001	2.560	1.303–5.030	0.006
Angiolymphatic invasion	4.708	2.487–8.914	<0.001	2.240	1.077–4.510	0.030

Note: Group 1: Total thyroidectomy without incidental lymph node resection; Group 2: Total thyroidectomy with incidental lymph node resection without lymph node metastasis; Group 3: Total thyroidectomy with incidental lymph node resection and lymph node metastasis; Group 4: Total thyroidectomy with central compartment neck dissection without lymph node metastasis; Group 5: Total thyroidectomy with therapeutic central compartment neck dissection (with lymph node metastasis). Variables such as nodal status and lymph node ratio (group dependent), macroscopic extra-thyroidal extension (pT status dependent) and ATA risk of recurrence stratification (multivariable dependent) were not included in these analyses. Legend: ^1^ Hazard ratio; ^2^ Confidence interval; ^3^ Cox proportional-hazards regression model; ^4^ According to American Joint Committee on Cancer Tumor Node Metastasis (AJCC-TNM) 8th edition; ETE—Extra-thyroidal extension.

**Table 6 cancers-15-00943-t006:** Exploratory characteristics of the patients from Group 3 (patients submitted to total thyroidectomy with incidental lymph node resection and lymph node metastasis) divided by treatment failure.

	No Failure: 25 (67.6%) ^1^	Treatment Failure: 12 (32.4%) ^1^	*p*–Value
Age			
Mean (SD)	42 (12)	44 (13)	0.909 *
Sex			
Male	5 (20%)	2 (17%)	0.594 **
Female	20 (80%)	10 (83%)	
WBS	24 (96%)	12 (100%)	0.676 **
RDIT	24 (96%)	11 (92%)	
Tumor size-largest foci (cm)			
Mean (SD)	1.79 (0.90)	2.21 (2.03)	0.871 *
Median (IQR)	1.50 (1.20, 2.00)	1.60 (0.88, 2.55)	
Range (min, max)	0.40, 4.00	0.30, 6.40	
Tumor multifocality ****	16 (64%)	10 (83%)	0.209 **
Microscopic ETE	16 (64%)	9 (75%)	0.391 **
Angiolymphatic invasion	11 (44%)	4 (33%)	0.835 **
pT status AJCC 8th ed.			
pT1a	3 (12%)	5 (42%)	0.0279 ***
pT1b	15 (60%)	2 (17%)	
pT2	4 (16%)	3 (25%)	
pT3a	-	1 (8.3%)	
pT3b	3 (12%)	-	
pT4a	-	1 (8.3%)	
Group staging AJCC 8th ed.			
I	24 (96%)	9 (75%)	0.129 ***
II	1 (4.0%)	2 (17%)	
III	-	1 (8.3%)	
Resected nodes			
Mean (SD)	3 (1)	3 (2)	0.538 *
Median (IQR)	2 (2. 4)	3 (3. 3)	
Range (min; max)	1; 6	1; 8	
1 node	5 (20%)	2 (17%)	0.137 ***
2 nodes	8 (32%)	1 (8.3%)	
3 nodes	4 (16%)	6 (50%)	
≥4 nodes	8 (32%)	3 (25.3%)	
Metastatic nodes			
Mean (SD)	1 (1)	2 (1)	0.629 *
Median (IQR)	1 (1. 2)	1 (1, 2)	
Range (min; max)	1; 4	1; 3	
1 node	17 (68%)	7 (58%)	0.767 ***
2 nodes	6 (24%)	4 (33%)	
3 nodes	1 (4%)	1 (8.3%)	
≥4 nodes	1 (4%)	-	
Lymph node ratio			
Mean (SD)	0.63 (0.32)	0.57 (0.27)	0.751 *
Median (IQR)	0.50 (0.33, 1.00)	0.58 (0.33, 0.69)	
Range (min, max)	0.17, 1.00	0.12, 1.00	
Node metastasis size (mm)			
Mean (SD)	1.54 (1.57)	3.2 (3.52)	0.151 *
Median (IQR)	0.8 (0.5, 6)	1.4 (0.57, 5.5)	
Range (min, max)	0.2, 6	0.3, 11	
LN macro metastasis (>2 mm)	8 (32%)	5 (41.7%)	0.413 **
Extranodal extension	2 (8%)	2 (16.7%)	0.391 **
PTC subtype			
Classic	21 (84%)	8 (67%)	0.278 ***
Follicular	2 (8.0%)	3 (25%)	
Oncocytic	1 (4.0%)	-	
Solid forms	-	1 (8.3%)	
Other	1 (4.0%)	-	
Risk of recurrence *****			
Low risk	2 (8%)	-	0.534 ***
Moderate risk	22 (88%)	11 (91.7%)	
High risk (min, max)	1 (4.0%)	1 (8.3%)	
Stimulated thyroglobulin (ng/mL)			
Mean (SD)	2.65 (3.66)	19 (24)	0.009 *
Median (IQR)	1.10 (0.25, 3.00)	7 (2, 31)	
Range (min, max)	0.20, 15.50	0, 80	
RDIT Dose (mCi)			
Mean (SD)	204 (17)	198 (43)	0.959 *
Median (IQR)	206 (202. 213)	206 (190. 209)	
Range (min, max)	150, 223	104, 266	
Follow-up (months)			
Mean (SD)	81 (21.29)	85 (35.18)	0.871 *
Median (IQR)	78 (66, 107)	80 (78, 99)	
Range (min, max)	2, 133	32, 112	

Legend: ^1^ n (%); AJCC—American Joint Committee on Cancer; WBS—Whole-body scintigraphy; RDIT—Radioiodine therapy; ETE—Extra thyroidal extension; PTC—papillary thyroid carcinoma; SD—standard deviation; SD—Standard deviation; IQR—Inter-quantile range; mCi—millicuries; * Mann–Whitney U test; ** Fisher’s exact test.; *** Chi-square test; **** Regarding the multiples tumor foci which may present within the thyroid gland as a result of either intrathyroidal tumor metastasis or coexistence of separate neoplastic foci; ***** According to ATA—American Thyroid Association 2015 guidelines.

## Data Availability

Not applicable.

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
