# Peer review of "Incidental Node Metastasis as an Independent Factor of Worse Disease-Free Survival in Patients with Papillary Thyroid Carcinoma"

_cancers, 2023, doi:10.3390/cancers15030943_

Round 1

Reviewer 1 Report

The simple summery is unclear, kept reading multiple time to identify the aim and rationale driving the study.

Line 37: greater tumor, is it larger tumors?

How incidental LN was defined, cN0 then discovered to be pN1?

The details of the patients' data are a comprehensively collected and well described. Statistical analysis was well done.

In addition to table 3, suggest plotting lab changes by time or at different time points, and compare the trends per groups.

Reviewer 2 Report

This manuscript discussed the the prognosis of PTC patients with incidental lymph node metastasis. The results showed that DFS was lower in patients pN1a-incidental compared to patients Nx and pN0-incidental but similar when compared to patients pN1a-CCND. In addition, multivariate analysis demonstrated that pN1a-incidental was an independent risk factors for lower DFS. There were some suggestions as following.

-There were 698 patients in this study, including 320 Nx, 264 pN0-incidental, 37 pN1a-incidental, 32 pN0-CCND and 45 pN1a-CCND. It means 301 PTC patients with incidental lymph node dissection. What is the incidental lymph node dissection rate?

-In Table1 (Tumor size), the minimum size is 0.01cm in several groups. How was such a tiny lesion found before surgery.

-In Table1, the rate of multifocality nearly 50%. whats the definition of multifocality.

-In results (section of 3.3), 156 patients performed RDIT in group 1. However, the moderate and high risk PTC patients were 82 in group 1 (Table 1). Why so many patient (62 cases) with low risk performed RDIT?

-In discussion, the author stated that lymph node size, foci size and extra-nodal extension is useful to stratify pN1a patients. The importance of extra-nodal extension has been emphasized in the latest study (PMID: 34482990) which should be cited in the manuscript.

-More than half of references beyong 5 yeas. If possible, update those references. 

Round 2

Reviewer 2 Report

The manuscript has been sufficiently improved to warrant publication in Cancers.